# The Effect of the Addition of Copper Particles in High-Density Recycled Polyethylene Matrices by Extrusion

**DOI:** 10.3390/polym14235220

**Published:** 2022-11-30

**Authors:** Camila Arcos, Lisa Muñoz, Deborah Cordova, Hugo Muñoz, Mariana Walter, Manuel I. Azócar, Ángel Leiva, Mamié Sancy, Gonzalo Rodríguez-Grau

**Affiliations:** 1Departamento de Ingeniería Mecánica y Metalúrgica, Facultad de Ingeniería, Pontificia Universidad Católica de Chile, Santiago 7820436, Chile; 2Instituto de Química, Facultad de Ciencias, Pontificia Universidad Católica de Valparaíso, Valparaíso 2373223, Chile; 3Departamento de Química de los Materiales, Facultad de Química y Biología, Universidad de Santiago de Chile, Santiago 7820436, Chile; 4Facultad de Química y Farmacia, Pontificia Universidad Católica de Chile, Santiago 7820436, Chile; 5Escuela de Construcción Civil, Facultad de Ingeniería, Pontificia Universidad Católica de Chile, Santiago 7820436, Chile; 6CIEN UC, Pontificia Universidad Católica de Chile, Santiago 7820436, Chile

**Keywords:** recycled polymers, high-density polyethylene, Fourier-transform infrared spectroscopy, differential scanning calorimetry, thermogravimetric, tensile testing, reflectance

## Abstract

In this study, the effect of the recycling process and copper particle incorporation on virgin and recycled pellet HDPE were investigated by thermo-chemical analysis, mechanical characterization, and antibacterial analysis. Copper particles were added to pellet HDPE, virgin and recycled, using a tabletop single screw extruder. Some copper particles, called copper nano-particles (Cu-NPs), had a spherical morphology and an average particle size near 20 nm. The others had a cubic morphology and an average particle size close to 300 nm, labeled copper nano-cubes (Cu-NCs). The thermo-chemical analysis revealed that the degree of crystallization was not influenced by the recycling process: 55.38 % for virgin HDPE and 56.01% for recycled HDPE. The degree of crystallization decreased with the addition of the copper particles. Possibly due to a modification in the structure, packaging organization, and crystalline ordering, the recycled HDPE reached a degree of crystallization close to 44.78% with 0.5 wt.% copper nano-particles and close to 36.57% for the recycled HDPE modified with 0.7 wt.% Cu-NCs. Tensile tests revealed a slight reduction in the tensile strength related to the recycling process, being close to 26 MPa for the virgin HDPE and 15.99 MPa for the recycled HDPE, which was improved by adding copper particles, which were near 25.39 MPa for 0.7 wt.% copper nano-cubes. Antibacterial analysis showed a reduction in the viability of *E. coli* in virgin HDPE samples, which was close to 8% for HDPE containing copper nano-particles and lower than 2% for HDPE having copper nano-cubes. In contrast, the recycled HDPE revealed viability close to 95% for HDPE with copper nano-particles and nearly 50% for HDPE with copper nano-cubes. The viability of *S. aureus* for HDPE was lower than containing copper nano-particles and copper nano-cubes, which increased dramatically close to 80% for recycled HDPE with copper nano-particles 80% and 75% with copper nano-cubes.

## 1. Introduction

Global polymer production is increasing yearly, reaching 335 million tons in 2016 [1] and 368 million tons in 2019 [2]. With this increasing trend, global production is estimated to reach 1.1 billion tons by 2050 [3]. In general, a large part of these plastics is discarded after a single use, generating a severe environmental problem [4,5] that could accumulate 12 billion tons of plastic waste in landfills and the environment by 2050 [3,6,7]. Furthermore, it is estimated that the ocean receives 8 million tons of plastic waste each year [8]. Unfortunately, according to the UN Environment Program, as of 2015, only 9% of the 9 billion tons of plastics produced worldwide were recycled, while 79% were accumulated in landfills or dumped directly into the environment [9,10]. Therefore, a significant volume of plastics can be recycled to increase their lifespan while at the same time minimizing the use of natural resources, toxic materials, and pollutant emissions derived from the production of new plastics. Such an effort would comply with the UN’s 12th objective of ensuring sustainable consumption and production patterns.

Thermoplastics have an exciting relationship between their mechanical properties and economic cost [11,12]. Polypropylene (PP) and polyethylene both have low and high densities (LDPE and HDPE) that stand out and are used in different applications, such as packaging, the automotive industry, construction [13,14,15], toys, pipes, and bottles [16,17,18].

The widespread presence of polymeric materials in everyday objects could represent an excellent industrial alternative in the fight against infections caused by pathogenic microorganisms that contribute to increasing mortality worldwide [19]. At least 700,000 people die annually due to diseases caused by drug-resistant bacteria [20]. Hospital surfaces and environments require permanent disinfection since microbial recolonization can occur rapidly after cleaning [21]. In addition, contaminated hands of medical personnel and inanimate surfaces are carriers of nosocomial infections [22]. Thus, an alternative that contributes to confronting and avoiding this type of infection can be the design and the generation of surfaces with antimicrobial properties that prevent their propagation. In this sense, thermoplastics are appropriate materials when solving this scientific and technological challenge [23,24], thanks to their ability to melt and mold while maintaining their chemical properties stable under an accessible range of temperatures [25], which is an advantage of their industrial use. The interest in modifying recycled polymeric matrices to give them better mechanical properties and to add an antimicrobial function is shared by several current industries [7,26]. This is because their doping with antibacterial agents provides materials with broad industrial applicability, which is an excellent opportunity to generate products that inhibit bacterial growth or viral permanence. Metallic particles such as silver (Ag) and copper (Cu) are known to have antimicrobial action and are effective against a broad spectrum of bacteria. Cu nanocomposites and antibacterial polymers have been used for intrauterine devices [14,27,28]. Therefore, it is essential to develop materials that act effectively as antimicrobial agents yet, at the same time, are non-toxic to the user and have mechanical properties compatible with use in social gathering spaces.

In this work, we study the effect of the addition of copper particles and the recycling process on extruded high-density polyethylene (HDPE) through the changes in their mechanical, physical-chemical, and antimicrobial properties. Copper particles were added due to their unique physical, chemical, and biological properties, such as high antibacterial activity [29] and low cost, which adds great potential in various applications, for instance, coatings, everyday objects, and construction components for living spaces [21,22]. The incorporation of the Cu nano-particles into the polymeric matrix was performed using a single-screw extruder [17,30,31,32] at high temperatures between 190 and 220 °C and at 30 rpm. This instrument is a productive, simple, safe, economical, and accessible technique for the industry [33] that enables the generation of virgin and recycled polymeric materials with copper particles of both nano- and micro-meter sizes.

## 2. Methodology

### 2.1. High-Density Polyethylene

Virgin high-density polyethylene (HDPE) was purchased from Filaments c.a (Toronto, Canada) with a 1.75 mm diameter, and the recycled HDPE (rHDPE) was provided as pellets by Bastias & Ibarra Reciclaje de Plásticos Enterprise (Santiago, Chile). rHDPE polymers were extruded between 400 and 450 kg/h using a cylindrical screw (SCM-440 55 Hr) with a diameter of 150 mm. Samples were cooled in water at room temperature held in a stainless-steel container. Then, both samples were pelletized using a 10 Horse-Power (HP) cutter equipped with a 24-piece blade (Fuji) and a steel guillotine.

### 2.2. Synthesis of Copper Particles

Cu-NPs were synthesized by mixing a solution of copper chloride (100 mL, 0.05 M) and 50 mL of ascorbic acid (0.1 M) at 80 °C while being stirred for 20 h (900 rpm). The suspension was filtered and stored at room temperature. Morphological characterization and high-magnification images (>5000) were acquired through TEM microscopy (Hitachi HT7700, Tokyo, Japan). The transmission electron microscopy (TEM) sample preparation was carried out by dipping Lacey carbon formvar-coated copper grids (300 Square, Pelco, CA, USA) in a diluted suspension (ethanol/H_2_O solution (3:1)) of samples. Then, the sample was air-dried before observation.

Cu-NCs were synthesized following a method previously reported by Gou et al. [34]. In brief, the copper (II) sulfate salt was reduced in an ascorbic acid solution in the presence of polyethylene glycol (PEG) and sodium hydroxide at 25 °C for 20 min. Later, the suspension was centrifugated, washed twice, and dried for 24 h at 70 °C. Cu-NCs were characterized by a scanning electron microscope (SEM, ZEISS, model EVO MA10, Jena, Germany) operated at 20 kV in a high vacuum.

Figure 1 reveals a spherical morphology for Cu-NPs, with an average particle size near 20 nm, and a cubic morphology for Cu-NCs, with a more significant average particle size close to 300 nm. Cu-NPs and Cu-NCs were first centrifuged for 1 min at 15,000 rpm. After that, the supernatant solvent was extracted, keeping the Cu-NPs and Cu-NCs moist. The precipitate was deposited and massed on Petri dishes.

### 2.3. Incorporation of Copper Particles on HDPE by Extrusion

Virgin and recycled HDPE samples were pelletized and mixed with Cu-NPs and Cu-NCs by extrusion, using a tabletop single screw extruder (Thera Instruments, BROX Spa, Santiago, Chile). HDPE samples were obtained with a concentration close to 0.5 wt.% Cu-NPs in HDPE and rHDPE and close to 0.7 wt.% Cu-NCs in HDPE and rHDPE. Figure 2 shows digital images of virgin and recycled HDPE samples.

### 2.4. Morphological Characterization

The morphology and size of the pure and modified samples were analyzed using an optical microscope (OM), model OLYMPUS BX60. QCapture Pro (version 6, Teledyne Imaging, TUS, USA) this software was used to perform OM image analysis to estimate the particle sizes. Furthermore, the HDPE samples were analyzed by field-emission scanning electron microscopy (FE-SEM, QUANTA 250 FEG, Hillsboro, OR, USA) which was equipped with an energy dispersion spectrometer.

### 2.5. Differential Scanning Calorimetry and Thermogravimetric Measurements 

The effect of the recycling process and the addition of Cu-NPs and Cu-NCs on HDPE samples were analyzed using differential scanning calorimetry (DSC). First, 12 to 16 mg of samples placed in an Al holder and sealed in a microcapsule by DSC (Perkin Elmer 6000, Waltham, MA, USA). The samples were heated to 280 °C, then cooled to 20 °C, and analyzed at a rate of 20 °C/min under a nitrogen atmosphere of 20 mL/min. Thermogravimetric analysis (TGA) was carried out using 10 mg of samples, which were exposed to a nitrogen atmosphere at 50 mL/min, starting at room temperature until reaching 800 °C at 10 °C/min, using a TGA 851e (Mettler-Toledo, Hong Kong, China). Both analyses were performed in duplicate.

### 2.6. Tensile Testing

The effect of the recycling process and the addition of Cu-NPs and Cu-NCs on HDPE were also investigated using a traction machine (Tinius & Olsen Mod. Super L, Shanghai, China), with a displacement speed of 50 mm/min, which was applied until the samples ruptured, as indicated in ASTM D638 [34]. Measurements were carried out five times each. Moreover, the hardness of the samples was measured using a durometer with a Shore D hardness scale, as described by the ASTM D 2240 [35] and ISO 868 standard [36]. The samples were molded using a hydraulic press (HP Industrial Instruments, Karnataka, India) under a pressure of 50 bar at 140 °C, which then were cooled under pressure with water circulation, obtaining 1 mm-thick samples, as shown in Figure 3. Each measurement was performed in triplicate.

### 2.7. Antibacterial Properties

Samples of virgin and recycled HDPE which had been modified with 0.5 wt.% Cu-NPs and 0.7 wt.% Cu-NCs were analyzed to evaluate their antibacterial potential against *Escherichia coli* BW 25,113 and *Staphylococcus aureus* ATCC 2538. The analysis was based on the E2180-01 [37] standard protocol, using Mueller Hinton agar and incubated at 25 °C, which was performed in duplicate.

## 3. Results and Discussion

### 3.1. Morphological Characterization

Figure 4 shows FE-SEM images of the HDPE and rHDPE samples modified with Cu-NPs and Cu-NCs, revealing the formation of cracks on the surface that had a lower density for samples modified with Cu-NPs. In addition, the rHDPE showed the incorporation of metal particles more clearly, with more homogenous dispersion for HDPE and rHDPE modified with Cu-NPs. Figure 4 also reveals an agglomerated distribution of Cu-NCs in rHDPE, with the size of the agglomerates being close to 4 µm, as reported by Gao et al. [38]. Figure 4 shows FE-SEM images of the HDPE with a higher magnification.

### 3.2. Differential Scanning Calorimetry and Thermogravimetric Measurements

Figure 5a–f shows the differential scanning calorimetry (DSC) heating and cooling curves of HDPE and rHDPE modified with Cu-NPs and Cu-NCs. From the DSC heating curves, the melting temperature ($T_{m}$) and melting enthalpy (${\Delta H}_{m}$) were obtained, and from the DSC cooling curves, the crystallization temperature ($T_{c}$) and the crystallization enthalpy (${\Delta H}_{c}$) were estimated, as also shown in Table 1 and Figure 5.

DSC curves revealed a slight reduction in $T_{m}$ with the recycling process, from 141.1 °C for HDPE to 138.1 °C for the rHDPE. For a virgin HDPE, Cuadri and Martín-Alfonso [39] reported a $T_{m}$ close to 133.8 ± 1 °C. Nevertheless, Muniyandi et al. [40] found a $T_{m}$ near 115.8 °C, which decreased to 108.2 °C with the recycling process. Therefore, a direct relationship could be established between the decrease in $T_{m}$ and the recycling process of HDPE. The incorporation of Cu-NPs in the virgin and recycled matrix of HDPE decreases the $T_{m}$ between 1 and 2 °C, which was more drastic with Cu-NCs, reducing by 3 °C. According to Gao et al. [38] the $T_{m}$ did not decrease their value with the addition of nanoparticles in the virgin HDPE matrix, such as graphene oxide [41], ZrO_2_ [42], titanate nanotubes (Y_2_W_3_O_12_) [41], SnO_2_ [43], and ZnO [38]. Therefore, the effect on $T_{m}$ due to incorporating Cu nanoparticles suggests an enhancement in the thermal properties of the HDPE matrix.

Based on the DSC analysis, the ${\Delta H}_{m}$ of the polymer was estimated by integrating the area under the endothermic curve (in J/g). This allowed us to estimate the degree of crystallization ($X_{c}$), according to equation 1 [44], where the enthalpy of fusion used for 100% HDPE crystallization was 293 J mol^−1^ [45]:(1)$$X_{c}=\frac{{\Delta H}_{m}}{\Delta H_{m}^{0}}\times 100\;\%$$

The $X_{c}$ was close to 49.7 and 42.4 for the HDPE and rHDPE samples, respectively, as shown in Table 1. This slight decrease can be attributed to the effect of the recycling process on the structure and high packaging organization [46,47,48,49]. Some researchers have proposed that the shear forces during the processing can induce the cleavage of the polymer backbone leading primarily to chain shortening due to the recycling process [50]. The $X_{c}$ also decreases for rHDPE with the addition of Cu-NPs and Cu-NCs compared to HDPE, in agreement with Jeziorska et al. [51,52], who previously reported that the incorporation of silica nanoparticles (SiO_2_) and silica-containing nano copper (Cu-SiO_2_) could decrease the crystallinity degree of the HDPE. For example, the $X_{c}$ for virgin HDPE was 62.5% that reduced to 53% by adding SiO_2_. The addition of Cu-SiO_2_ allowed us to obtain 59.9% of crystallinity [50,51]. In contrast, the $X_{c}$ increases for HDPE with Cu-NPs and Cu-NCS are shown in Table 1. Similarly, Gao et al. [38] reported that the addition of ZnO nano-particles increases the $X_{c}$ in concentrations between 1 and 5 wt. %. Mahmoud et al. [53] found that when the content of nano-particles increased in the polymer with a uniform distribution, the crystalline regions increase, causing higher crystallinity in HDPE.

Figure 6 shows the thermogravimetry analysis (TGA) for HDPE and rHDPE samples, both modified with Cu-NPs and Cu-NCs, revealing that the recycling process influenced the initial temperature of HDPE. Thermal degradation is associated with the recycling process, which can be attributed to a change in the molecular weight without forming volatile products, resulting from the cleavage of weak links, such as oxygen bridges and lateral chains [20]. Several authors have reported that the heating applied during the recycling and pelletizing process can favor the formation of shorter chains in the HDPE samples, which are less thermally stable than virgin material [46,51,54]. Figure 6 also shows minor differences in the thermal degradation profiles, especially in the maximum thermal degradation rate (minimum of the curve), which is slightly lower for recycled polymers with Cu-NPs than virgin polymers with Cu NPs. The slight differences are possibly related to the content of Cu as NPs and NCs in each sample.

### 3.3. Mechanical Characterization 

It has been reported that the mechanical properties decrease for high loading of the HDPE matrix’s nano-filler [55]. Therefore, the challenge was to improve the mechanical properties without destroying the other polymeric characteristics. In this context, the effect of the recycling process and incorporation of Cu-NPs and Cu-NCs on the mechanical behavior of HDPE was studied using stress–strain curves and hardness measurements, as shown in Figure 7.

Figure 7 reveals a drastic effect on the mechanical properties of the HDPE from the recycling process, such as the maximum tensile strength $\left(\sigma_{\max}\right)$, the tensile strength at break ($\sigma_{B}$), elongation at break ($e_{B}$), and shore D hardness (D), as also shown in Table 2. The properties of the virgin HDPE were degraded by the recycling process but enhanced by the addition of Cu particles, which can be associated with a modification of complex polymeric chains during the tension processes [54,56]. Figure 7 and Table 2 show that the maximum strength $\sigma_{\max}$ decreased slightly with the recycling process compared to the virgin sample that increased with the addition of Cu-NPs and Cu-NCs. The copper particles could change the crystallinity of samples, possibly enhancing the defects, such as vacancies and dislocations. Furthermore, adding Cu-NPs and Cu-NCs into the virgin and recycled HDPE increases the Yield strength and shows a paradoxical effect on the strain, growing slightly in the virgin HDPE samples and decreasing mildly in the recycled HDPE.

Singh et al. [56] studied the mechanical properties of a polymeric matrix composed of (50/50) virgin and recycled HDPE, obtaining a $\sigma_{\max}$ value of 12.27 MPa, which reduced the $\sigma_{\max}$ value of virgin HDPE by 30%, which was close to 17.39 MPa. Figure 7 and Table 2 show that the $\sigma_{B}$ of virgin HDPE decreased almost four times with the addition of Cu-NPs and Cu-NCs in the samples. The $e_{B}$ of the polymeric matrix increased with the recycling process and incorporation of copper particles, in some cases more than the double. Tesfaw et al. [57] determined a tension of 20.67 MPa and an elongation of 25.13 mm for a virgin HDPE and a tension of 7.73 MPa and 9.97 mm of elongation for r-HDPE; Beigloo et al. [55] also reported a similar effect. The increase in the elongation suggests that the copper particles could increase the ductility of HDPE, changing the crystalline/amorphous zones or increasing the local and linear defects. Table 2 also shows that the shore D hardness was slightly reduced with the recycling process and addition of Cu-NCs, which was more drastic with Cu-NPs; therefore, the size particle and/or morphology slightly influence the mechanical properties of HDPE.

Therefore, the incorporation of nano-particles improved the mechanical responses of recycled and virgin HDPE matrices, which can be associated with the high surface areas of the nano-particles [56,58,59,60], which can influence the movement of defects and polymer chains [61,62], even with a 0.5% Cu-NP content and a 0.7% Cu-NC content.

### 3.4. Antibacterial Properties

Figure 8 shows the bacterial viability of virgin and recycled HDPE samples exposed to a medium inoculated with *E. coli* and *S. aureus*. The virgin HDPE showed a reduction in the viability of *E. coli* in samples containing Cu-NPs and Cu-NCs, the latter revealing the lowest percentage of viability. The virgin HDPE sample exposed to a medium inoculated with *S. aureus* also showed that the viability of the bacteria was lower in the samples with Cu-NPs and Cu-NCs, the latter having the lowest percentage of viability, being the best in terms of antibacterial activity. In addition, the recycled polymer showed that the best antibacterial activity was obtained by the sample containing Cu-NCs. The rHDPE sample showed much higher viability percentages, as shown in Appendix A. Furthermore, the different effects of the Cu-NPs and Cu-NCs can be attributed to their dispersion in the polymeric matrix, as shown in Figure 4.

Vásquez et al. [63] studied the antibacterial effect of incorporating ZnO-NPs in recycled polyethylene terephthalate (r-PET) polymeric fibers, and they measured the diameter of inhibition (in mm) for the bacteria *E. coli* and *S. aureus*. The antibacterial properties of r-PET were lower than virgin PET, using the same quantity of added ZnO-NPs, suggesting that the recycling process affects the antibacterial behavior of the polymers. However, only the antibacterial properties of virgin HDPE with the Cu-NPs and nano-fibers have been reported [17,64,65,66]. The authors reported a decrease in CFU (colony-forming units) by increasing the concentration of NPs in the polymeric matrix, which was associated with the release of copper ions that can penetrate the membrane of the bacteria and affect their viability. Shankar et al. [65], Arfat et al. [66], and Chatterjee et al. [67] proposed that the antibacterial effect of Cu-NPs was related to its toxicity to reactive oxygen species. Tamayo et al. [17] reported that the antibacterial behavior depends on three factors: the synergy between the polymer and the Cu-NPs, the capacity of the polymer for long-term ion release, and the dispersion of the Cu-NPs in the polymer matrix.

## 4. Conclusions

In this work, the effect of Cu particles on HDPE was studied by a surface analysis, thermo-chemical analysis, mechanical tests, and viability essays.

DSC analysis revealed that the recycling process decreased the melting temperature, being 139.57 °C for virgin HDPE and 132.6 °C for the recycled HDPE. Nevertheless, incorporating Cu-NPs and Cu-NCs increased the melting temperature for the recycled HDPE, reaching almost the melting temperature of a virgin HDPE, being 134.67 °C for recycled HDPE with Cu-NPs, and 136.36 °C for recycled HDPE with Cu-NCs.

Thermogravimetric analysis revealed that the recycling process influenced the initial decomposition temperature, reaching 420 °C for virgin HDPE and 398 °C for the recycled HDPE. This effect was associated with the heating applied during the recycling and pelletizing process. However, the decomposition temperature was not modified at 50 %, being close to 494 °C for virgin HDPE and 490 °C for recycled HDPE. A similar response was determined at 100%, near 524 °C for virgin HDPE and 518 °C for recycled HDPE.

The tensile tests revealed that the properties of the virgin HDPE were degraded by the recycling process but enhanced by adding Cu particles, which can be associated with a modification of complex polymeric chains during the tension processes. Nevertheless, the addition of copper particles increased the tensile strength of the recycled polymers. For instance, the maximum tensile strength was nearly 26.97 MPa for the virgin HDPE, which decreased to 21.51 MPa for the recycled HDPE. However, it was nearly 24.38 MPa with the addition of 0.5 wt.% Cu-NPs and close to 25.39 MPa with the incorporation of 0.7 wt.% Cu-NCs. In addition, the elongation at break increased with the incorporation of copper particles. For example, it was near 104.80 % for the recycled HDPE, 62.41 % with 0.5 wt.% Cu-NPs, and close to 105.87 % with 0.7 wt.% Cu-NCs.

A reduction in the viability for *E. coli* of 8% was seen in virgin HDPE samples containing Cu-NPs, and the viability in samples with Cu-NCs was lower than 2%. The recycled HDPE samples showed much higher viability percentages, being close to 95% with Cu-NPs and 45% with Cu-NCs. The viability of *S. aureus* for HDPE with Cu-NPs was lower than 2% and less than 1% with Cu-NCs. However, it was nearly 80% for the recycled HDPE samples with Cu-NPs and close to 75% Cu-NCs. Therefore, the antibacterial activity of the recycled polymer was improved with the addition of copper particles, mainly with the sample containing Cu-NCs, which has a more substantial antibacterial effect against *E. coli*.

## Figures and Tables

**Figure 1 polymers-14-05220-f001:**
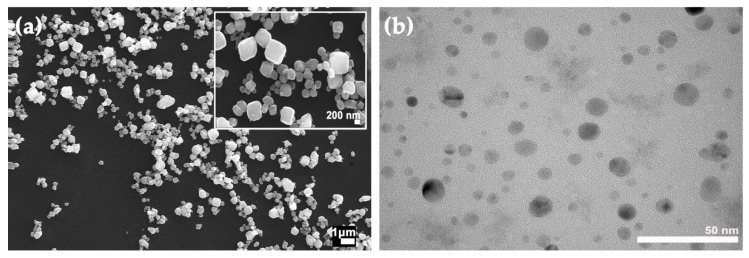
(**a**) FE-SEM images of Cu-CPs and (**b**) TEM images of Cu-NPs.

**Figure 2 polymers-14-05220-f002:**
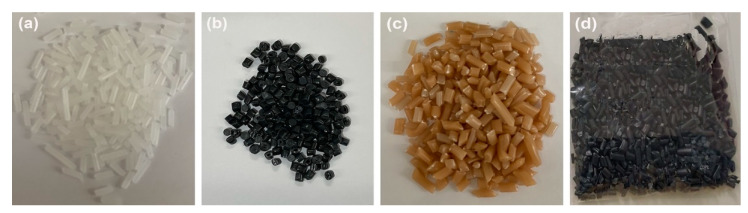
Images of samples (**a**) HDPE, (**b**) rHDPE, (**c**) HDPE–Cu-CPs, and (**d**) rHDPE–Cu-CPs.

**Figure 3 polymers-14-05220-f003:**
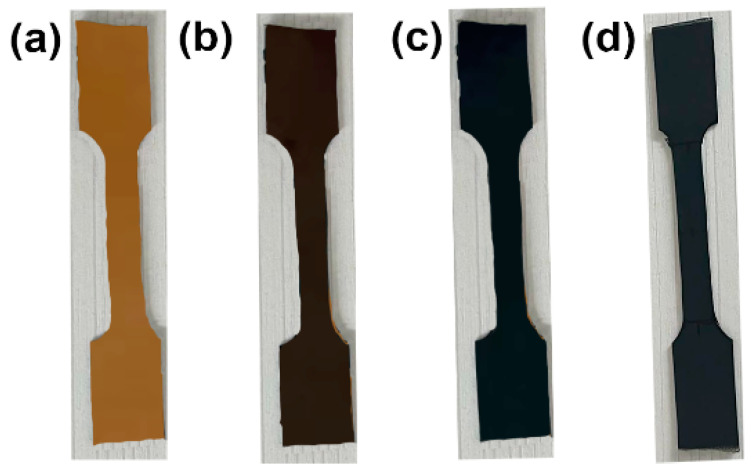
Digital images of mechanical samples of (**a**) HDPE 0.7 wt.% Cu-NCs, (**b**) HDPE 0.5 wt.% Cu-NPs, (**c**) rHDPE 0.7 wt.% Cu-NCs, and (**d**) rHDPE 0.5 wt.% Cu-PCs.

**Figure 4 polymers-14-05220-f004:**
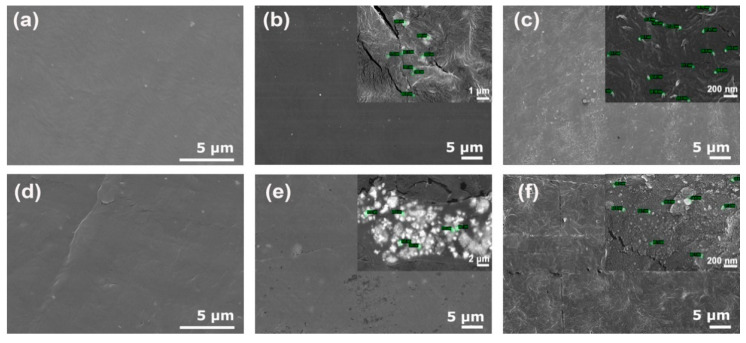
FE-SEM images of (**a**) HDPE, (**b**) HDPE 0.7 wt.% Cu-NCs, (**c**) HDPE 0.5 wt.% Cu-NPs, (**d**) rHDPE, (**e**) rHDPE 0.7 wt.% Cu-NCs, and (**f**) rHDPE 0.5 wt.% Cu-NPs.

**Figure 5 polymers-14-05220-f005:**
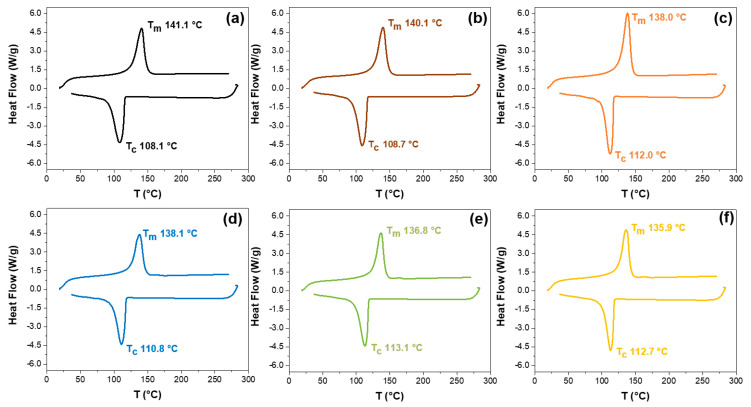
Effect of recycling and addition of Cu-NPs on the differential scanning calorimetry of (**a**) HDPE, (**b**) HDPE 0.5 wt.% Cu-NPs, (**c**) HDPE 0.7 wt.% Cu-NCs, (**d**) rHDPE, (**e**) rHDPE 0.5 wt.% Cu-NPS, and (**f**) rHDPE 0.7 wt.% Cu-NCs.

**Figure 6 polymers-14-05220-f006:**
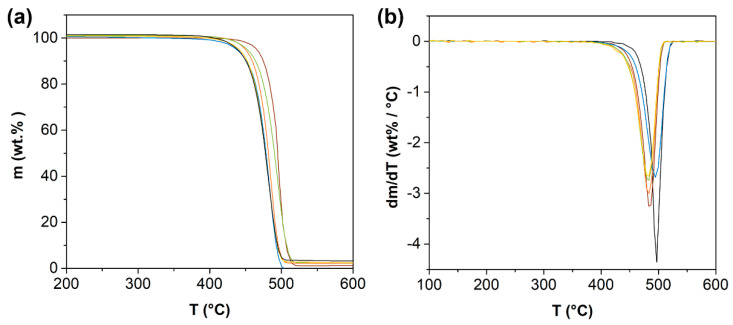
Effect of the recycling and the addition of Cu-NPs and Cu-NCs on the thermogravimetry of (

) HDPE, (

) HDPE 0.5 wt.% Cu-NPs, (

) HDPE 0.7 wt.% Cu-NCs, (

) rHDPE, (

) rHDPE 0.5 wt.% Cu-NPs, and (

) rHDPE 0.7 wt.% Cu-NCs. (**a**) TGA analysis and (**b**) maximum thermal degradation rate.

**Figure 7 polymers-14-05220-f007:**
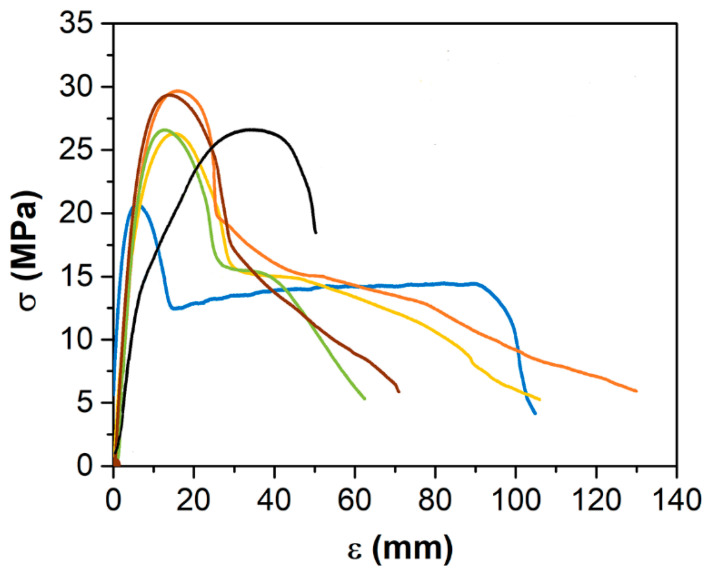
Effect of the recycling and the addition of Cu-NPs and Cu-NCs on the mechanical properties of (

) HDPE, (

) HDPE 0.5 wt.% Cu-NPs, (

) HDPE 0.7 wt.% Cu-NCs, (

) rHDPE, (

) rHDPE 0.5 wt.% Cu-NPs, and (

) rHDPE 0.7 wt.% Cu-NCs.

**Figure 8 polymers-14-05220-f008:**
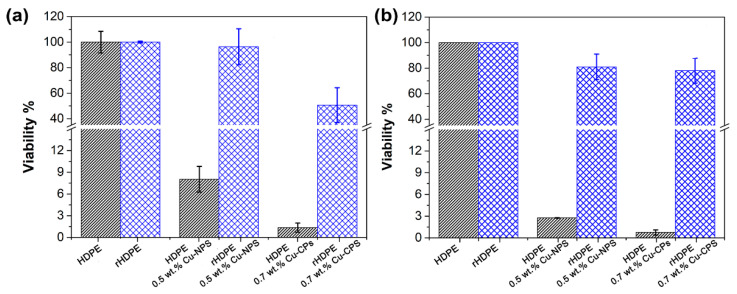
Comparison of the bacterial viability of (
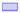
) virgin HDPE and (
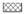
) recycled HDPE samples: (**a**) *E. coli* and (**b**) *S. aureus*.

**Table 1 polymers-14-05220-t001:** Differential scanning calorimetry parameters for HDPE.

Samples	$T_{m}\;\left({}^{\circ}C\right)$	$T_{c}\;\left({}^{\circ}C\right)$	$\Delta H_{m\;}\left(J/g\right)$	ΔHc (J/g)	$X_{c}\;\left(\%\right)$
HDPE	141.1	108.1	145.5	159.6	49.7
rHDPE	138.1	110.8	124.1	148.9	42.4
HDPE-0.5 wt.% Cu-NPs	140.1	108.7	165.7	166.4	56.6
rHDPE-0.5 wt.% Cu-NPs	136.1	113.1	143.2	144.6	48.9
HDPE-0.7 wt.% Cu-NCs	138.0	112.0	198.6	148.3	67.8
rHDPE-0.7 wt.% Cu-NCs	135.9	112.7	144.2	140.7	49.2

**Table 2 polymers-14-05220-t002:** Mechanical parameters estimated from stress–strain curves.

Samples	$\sigma_{max}\;(\mathrm{MPa})$	$\sigma_{B}\;(\mathrm{MPa})$	$e_{B}\;(\mathrm{mm})$	D (xHD)
HDPE	26.97± 0.5	18.86 ± 0.3	52 ± 1.89	59.3 ± 0.8
rHDPE	21.51 ± 0.9	4.46 ± 0.5	259.9 ± 43.5	57.9 ± 1.6
HDPE-0.5 wt.% Cu-NPs	28.03 ± 1.4	5.58 ± 0.3	126.7 ± 30.6	55.4 ± 2.1
rHDPE-0.5 wt.% Cu-NPs	26.43 ± 0.2	5.28 ± 0.04	140 ± 28.3	54.6 ± 2.8
HDPE-0.7 wt.% Cu-NCs	29.58 ± 1.2	6.25 ± 0.4	170 ± 14.1	57.2 ± 2.1
rHDPE-0.7 wt.% Cu-NCs	25.09 ± 1.6	5.02 ± 0.3	166.7 ± 46.2	57.6 ± 1.3

## Data Availability

The raw/processed data required to reproduce these findings cannot be shared at this time due to technical or time limitations.

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
