# Peer review of "The Effect of the Addition of Copper Particles in High-Density Recycled Polyethylene Matrices by Extrusion"

_polymers, 2022, doi:10.3390/polym14235220_

Round 1

Reviewer 1 Report

This paper reports the properties of high-density polyethylene (HDPE) and recycled HDPE (rHDPE) containing Cu nano particles and cubes. The following issues must be resolved before this paper is published.

1. Abbreviations, such as Cu-NPs and Cu-NCs in the Abstract; HP and pc in line 93, must be spelled out or explained at the first appearance.

2. The unit for 150 is missing in line 101.

3. Sample preparation methods or used instruments for FE-SEM and TEM are not explained.

4. The letter y in line 107 does not make sense.

5. The methods of thermal analyses using DSC are not explained sufficiently. The data in Figure 5 seems to have been obtained in the heating process, whereas a scan rate of 20 K/min was indicated for the cooling process. The heating rate is not indicated in the paper. The authors should also explain that the accuracy of five digits for temperature (line 166) is reliable enough.

6. The tensile mechanical properties in Figure 6 and Table 3 are compared with those in the literature for HDPE, whose values indicated in the Table vary widely. When they are compared with the values of the higher limit, adding Cu particles makes the properties worse. The authors must obtain these data to confirm that Cu particles improve mechanical properties.

7. The sample codes in Figure 7 are different from those in the text.

8. The word DCS in line 298 must have been misspelled.

Author Response

Dear Reviewer,

We appreciate your review of our manuscript. The comments received have been responded to in the manuscript and also in the attached reply file. 

Kind regards,
Gonzalo Rodríguez-Grau

Reviewer 2 Report

The following questions have to be answered:

Chapter 2.1: After reading the paragraph I have no idea which conditions was used for the processing? (Temperature, geometry design of the screw, pressure, etc) This is necessary for understanding the material behaviour....Also Figure 1 a and b are missing....

Chapter 2.2: I am confused...you were preparing Cu-particles out of a solution and create nano- and microsized particles - how you seperate them?

Chapter 2.3: I) The processing conditions are missing (see above). II) Why you use different filling contents for nano and microplastics? Is there any specific reason? The picture should be figure 2...

Chapter 2.4 I) How was the sample preparation? I have no idea about that....on the supplementory data I found a lot of bubbles inside....II) The SEM pictures look like a chemical degradation (mud crack pattern) - I am wondering why you cant find this with other methods....

Chapter 2.6 How you have prepared the samples? The quality of the samples look inadequat....How many specimen were prepared?

Chapter 3.1 see above... Figure 4 is named SEM and DSC-results - please check that

Chapter 3.2

DSC: How many measurements were perfomed? The curves in figure 5 are not looking very well....The peaks itself have often more than one peak - it seems to be that there is a second one below...(shoulder). What do the authors think about this phenomenom? However, the big question is the reproducability....otherwise the Tm seems to be very similar....

TGA: I had a look on the suppl. data and I cannaot found any difference in the Ti....please explain this in detail...Why the copper filled samples were not measured with this method?

Chapter 3.3

For me it is not clear howmany specimen were tested for detecting tensile properties? Why you are not presenting a stadard deviation? Why you have not evaluated the Youngs Modulus? The dedection of the breaking properties are definitivly wrong... The statement in line 230 is not clear....Why you did not measured th mechanical properties of the virgin HDPE? Why are these values only taken out of some publications?

Author Response

(The authors gave the same response as above.)

Reviewer 3 Report

Dear Authors,

I studied your manuscript entitled "The effect of the addition of copper particles in high-density recycled polyethylene matrices by extrusion". There are several weaknesses in the paper, resulting from the lack of novelty and poor broader impact (for example, readership and technical significance). Some spaces need to be improved in terms of journal quality. Thus, I do not recommend the publication of your hastily written paper in the Polymers journal without a deep revision.

1) The quality of the abstract and conclusion should be enhanced by the inclusion of significant research findings. More quantitative data in these sections would be beneficial.

2) How did you select the concentration of Cu (0.5 and 0.7 wt.%)? More details are needed about the HDPE materials (especially the melt flow index).

3) HDPE could be easily thermally and mechanically degraded during sample preparation. Its properties are affected by these treatments. Can you assume (with experimental support) that their results are free from these problems?

4) The title of sections 2.5 and 3.2 should be changed to "thermal analyses" or “differential scanning calorimetry and thermogravimetric measurements”.

5) Figure 5: The legend of the Y-axis should be mentioned. Please present and discuss the cooling curves.

6) Figure7: Error bars are needed and the number of replicates needs to be stated. What is the Cu concentration?

7) Table 2: Please present the TGA results for samples containing Cu particles.

8) Table 3: Standard deviations should also be considered.

9) The manuscript needs to be thoroughly revised because it contains a few typos and errors.

Author Response

(The authors gave the same response as above.)

Round 2

Reviewer 1 Report

The reviewer recommends adding parenthesized descriptions as follows, and use these abbreviations thereafter.

"called copper nano-particles (Cu-NPs)" (line 22)

"labeled copper nano-cubes (Cu-NCs)" (line 24)

Author Response

(The authors gave the same response as above.)

Reviewer 2 Report

The authors have included most of the comments. There is only one question from my side left:

Mechanical properties: How does the Young's Modulus (indicator for the stiffness of your material) fit to your explanations?

Additionally I found some small mistakes:

At line 226: You mean rHDPE instead of HDPEr...

Figure 6 in the figure desciption: The lines are shifted within the text...

Author Response

(The authors gave the same response as above.)

Reviewer 3 Report

Dear Authors,

Thank you for considering my comments. I have recommended the publication of your paper as is.

Author Response

Dear,

We greatly appreciate your comments and the good reception of the results presented.

best regards
